Adaptation of a clinical reasoning model for use in inflammatory conditions of the lactating breast: a retrospective mixed-methods study

Heron Emma 1 emma.duff@postgrad.curtin.edu.au
http://orcid.org/0000-0002-8464-6479 McArdle Adelle 2
Cooper Melinda 3
Geddes Donna 4
http://orcid.org/0000-0003-3007-8231 McKenna Leanda 1
1 School of Allied Health, Curtin University , Bentley, Western Australia , Australia
2 Monash Rural Health, Monash University , Churchill, Victoria , Australia
3 MMC Physiotherapy , Lauriston, Victoria , Australia
4 School of Molecular Sciences, University of Western Australia , Crawley, Western Australia , Australia
Prazeres Filipe
Electronic publication date: 2022 Jul 25
Publication date: 2022
Volume: 10
Electronic Location ID: e13627
Received 2021 Dec 23; Accepted 2022 Jun 2
Copyright: © 2022 Heron et al.
Copyright year: 2022
Copyright holder: Heron et al.
License: This is an open access article distributed under the terms of the Creative Commons Attribution License, which permits unrestricted use, distribution, reproduction and adaptation in any medium and for any purpose provided that it is properly attributed. For attribution, the original author(s), title, publication source (PeerJ) and either DOI or URL of the article must be cited.
License URL: https://creativecommons.org/licenses/by/4.0/

Keywords: Breastfeeding, Mastitis, Lactation disorders, Mothers, Physical therapists, Nipple, Decision making, Pain, Treatment, Assessment

Funding: Australian Government Research Training Program Scholarship Emma Heron undertook this work as part of the completion of a PhD and was supported by an Australian Government Research Training Program Scholarship (https://www.dese.gov.au/research-block-grants/research-training-program). The funders had no role in study design, data collection and analysis, decision to publish, or preparation of the manuscript.

==============================
Background:

Many potential factors associated with Inflammatory Conditions of the Lactating Breast (ICLB) have been reported in the literature, by lactating mothers and clinicians. Clinicians, including general practitioners, lactation consultants and physiotherapists, require a clinical reasoning model that summarises associated or linked factors, to aid in the assessment, treatment, and prevention of ICLB. Thus, we aimed to adapt the existing Breastfeeding Pain Reasoning Model (BPRM), for use in the management of ICLB, using prior research and clinical audit data to guide adaptation. The existing BPRM categorises contributing factors for breastfeeding nipple pain, rather than ICLB.

Methods:

Factors linked with ICLB were identified from prior research and considered for inclusion into the existing model. Clinical data from a retrospective audit of ICLB patient notes at a private physiotherapy practice were also examined. Any factors identified from prior research that could not be identified in the clinical notes were not considered for inclusion into the existing model. Additional factors from the clinical notes that appeared repeatedly were considered for inclusion into the adaptation of the BPRM. A draft adapted model was created comprising all eligible factors, considering their counts and percentages as calculated from the clinical data. The research team iteratively examined all factors for appropriate categorisation and modification within the adapted model.

Results:

Prior research and data from 160 clinical notes were used to identify factors for inclusion in the adapted model. A total of 57 factors, 13 pre-existing in the BPRM and 44 extra identified from the prior research or clinical audit, comprised the draft adapted model. Factor consolidation and terminology modification resulted in a total of 34 factors in the final proposed adapted ICLB model. The three main categories, CNS modulation, External influences and Local stimulation, from the existing model were maintained, with one minor terminology change to the former Local stimulation category, resulting in ‘Local influences’ category. Terminology for five subcategories were modified to better reflect the types of factors for ICLB. The most common factors in the adapted model, calculated from the clinical audit population of mothers with ICLB, were employment (85%), high socioeconomic status (81%), antibiotic use during breastfeeding (61%), history of an ICLB (56%), any breast pump use (45%), multiparity (43%), birth interventions (35%), decreased milk transfer (33%), breastfeeding behaviour and practices (33%), nipple pain (30%) and fit and hold (attachment and positioning) difficulty (28%).

Conclusion:

An ICLB-specific linked factors model is proposed in this paper. Clinicians treating mothers with ICLB can use this model to identify influencing and determining factors of ICLB clinical presentations and provide targeted education and effective treatment plans.

Introduction

Inflammatory Conditions of the Lactating Breast (ICLB) is an overarching term used by researchers and health care professionals to encompass the suite of breast conditions that present with local, and often systemic, inflammatory signs and symptoms during lactation (Cooper, Lowe & McArdle, 2020). Most Australian physiotherapy clinicians consider blocked ducts, engorgement, mastitis and breast abscess, as ICLB (Heron et al., 2020), with diagnosis based on maternal clinical signs and symptoms (Collado et al., 2009; Heron et al., 2020). Symptom severity is primarily driven by inflammatory mediators rather than pathogenic bacteria (Ingman, Glynn & Hutchinson, 2014; Kvist et al., 2008; Osterman & Rahm, 2000). Approximately one in four lactating mothers experience mastitis, or breast inflammation (Amir & The Academy of Breastfeeding Medicine Protocol Committee, 2014), within the first six months postpartum (Wilson, Woodd & Benova, 2020), and it is a commonly reported reason for prematurely ceasing lactation (Aktimur et al., 2016; Sun et al., 2017; Wockel et al., 2008).

Management and prevention strategies are often focused on addressing the potential factors involved in the initiation or prognosis of the ICLB (Amir & The Academy of Breastfeeding Medicine Protocol Committee, 2014). There is growing evidence that environmental and genetic factors can trigger breast inflammation without obvious increases in potentially pathogenic bacteria (Ingman, Glynn & Hutchinson, 2014). An exploration of potential factors linked to ICLB is important to assist in prioritisation and targeted management for lactating mothers with ICLB, as well as educating healthy lactating mothers. Additionally, an exploration of potential factors linked to ICLB is an important component to ensuring research correctly identifies and accounts for these factors to develop evidence-based treatments for ICLB.

While numerous potential factors linked to the development of ICLB have been reported in the literature, the evidence is poor and inconclusive (Amir & The Academy of Breastfeeding Medicine Protocol Committee, 2014; Wilson, Woodd & Benova, 2020). A recent systematic review of factors associated with lactational mastitis (Wilson, Woodd & Benova, 2020) identified 42 factors, however meta-analysis was impeded by methodological heterogeneity. A recent large case control study (n = 652 participants) (Yin et al., 2020), not included in the systematic review, investigated the association of 10 additional factors with lactational mastitis. Further, a meta-analysis on maternal risk factors for lactational mastitis (Deng et al., 2021), found 13 significant factors, however methodological issues, such as those raised previously (Senn, 2009), may have confounded findings. The overall high number of associated factors and inconsistencies across the studies can be confusing for clinicians to determine the potential influencing or contributing factors that form the foundation for the development of an appropriate individualised management plan. Similarly for researchers, this may prove difficult when trying to minimise or reduce the impact of confounding variables in interventional trials.

Our study aimed to adapt an existing clinical reasoning model for nipple pain associated with breastfeeding, the ‘Breastfeeding Pain Reasoning Model’ (Amir, Jones & Buck, 2015), for categorisation of ICLB linked factors to assist in conceptualisation for clinicians and researchers. The existing model was originally adapted from a clinical reasoning model developed for musculoskeletal pain, the ‘Pain and Movement Reasoning Model’ (Jones & O’Shaughnessy, 2014). These prior models used three integrated categories, central nervous system (CNS) modulation, regional or external influences and local stimulation, to capture the factors contributing to an individual’s pain experience (Amir, Jones & Buck, 2015; Jones & O’Shaughnessy, 2014). Similar to these prior models, our adapted model aims to help clinicians capture the influences and determinants of a mother’s ICLB episode, using a biopsychosocial categorisation approach. This will enable clinicians to identify and prioritise management strategies and assist in preventing or reducing ICLB recurrence. For researchers, our model aims to help target prioritisation of future association testing studies. Thus, the specific objective of this study was to summarise the factors linked with ICLB in an adapted model to facilitate and enhance clinical management of ICLB.

Materials and Methods

Design

This was a mixed methods study, using quantitative and qualitative data to adapt an existing clinical framework for use in patients who have ICLB, rather than nipple pain only. Ethics approval was obtained from Curtin University Human Research Ethics Committee (HRE2020-0544) for the retrospective clinical audit which supplied the quantitative data. Given the retrospective study design, a waiver of informed consent was granted, in accordance with section 2.3.10 of the National Statement on Ethical Conduct in Human Research (National Health & Medical Research Council, 2018).

The stages of development of the ‘ICLB reasoning model’ are outlined in Fig. 1 and described in detail below.

Figure 1 Stages of development of the adapted clinical reasoning model.

T, total number of factors for consideration.

Phase 1: model selection

The ‘Breastfeeding Pain Reasoning Model’ for nipple pain (Amir, Jones & Buck, 2015) was chosen due to its breastfeeding context and clinical suitability and utility (Jones et al., 2021). This existing model uses 25 factors, organised into three categories, and nine subcategories, that were focused on nipple pain only and not on ICLB (factors summarised in Table S1).

Phase 2: prior research factors

Five electronic databases (Medline (Ovid, New York, NY, USA), CINAHL Plus with full text (EBSCO, Ipswich, MA, USA), EMBASE (Ovid, New York, NY, USA), ProQuest and Scopus) were searched using the keywords ‘lactation’, ‘mastitis’ and ‘risk factors’. Results were limited to humans, English language, and publication year 2017 to 2021 given the systematic search performed by authors of the recent systematic review (Wilson, Woodd & Benova, 2020). Results were exported to EndNote, duplicates deleted, and the remaining studies searched for all investigated risk factors by one author (EH).

Factors associated with ICLB were extracted from the resulting prior research; a 2020 narrative synthesis on risk factors for lactational mastitis (Wilson, Woodd & Benova, 2020), a case-control study examining the risks for lactational mastitis (Yin et al., 2020) and a meta-analysis of maternal risk factors for lactational mastitis (Deng et al., 2021) (EH). These factors were cross-matched with the factors in the existing, ‘Breastfeeding Pain Reasoning Model’. Any factors identified in the prior research that were not present in the existing ‘Breastfeeding Pain Reasoning Model’ for nipple pain were considered for inclusion in the adapted model under the relevant category and subcategory. Factors that were already contained within the existing ‘Breastfeeding Pain Reasoning Model’ for nipple pain, remained for potential inclusion in the adapted model. Phase two was completed by one auditor (EH).

Phase 3: clinical audit factors

A retrospective audit of clinical notes of mothers with ICLB (n = 160) was undertaken by the authors as part of a companion study that examined the validity of an ICLB-specific patient reported outcome measure (Heron et al., 2021). For further details of the audit see the published companion study (Heron et al., 2021).

Participants

For the present study, the clinical notes were drawn from mothers who had presented to a private physiotherapy practice in Melbourne, Victoria from the 12th July 2017 to 15th September 2020 for treatment of an ICLB. The auditor (EH) worked chronologically backwards (from the 15th September 2020) in the practice management software, searching for all lactation appointments in the daily diary and screening corresponding clinician notes, until 160 eligible clinical notes were obtained. A hospital setting was not included as ICLB is less prevalent in the first three days postpartum, which is the usual duration before mothers have been discharged from hospital (LactaResearch Group, 2018). Australian mothers commonly seek treatment for ICLB from private clinics, where physiotherapists are primary contact practitioners (Diepeveen et al., 2019). Private healthcare is accessible for many Australians, especially those in metropolitan areas. Mothers usually live within 20 to 30 min of a clinic; most can drive, and many have private health insurance which offers a partial rebate of consultation fees (Diepeveen et al., 2019). The physiotherapy practice was chosen as it was a leading practice for ICLB treatment in Victoria, hosting the Australian ‘Lactation for Health Professionals’ course (Inform Physiotherapy, 2020), and it indicated that it had the required number of clinical notes.

For inclusion, clinical notes had to meet criteria for the companion validity study (Heron et al., 2021). This required the mother to be over 18 and the clinical notes to be complete and contain the full assessment items in the initial ICLB appointment. Incomplete clinical notes and repeat initial appointments were excluded.

Instrumentation/outcome measures

The eligible clinical notes were searched for the presence of all prior factors identified in phase 2 from the research (54). In addition, the clinical notes were searched for any documented factors that were potentially linked with the onset or aggravation of the mother’s ICLB episode or symptoms. These could have been patient reported and/or clinician confirmed. Factors that were identified in phase 2 or already contained within the ‘Breastfeeding Pain Reasoning Model’ for nipple pain, that were not identified in the clinical notes were not considered for inclusion in the adapted model.

The clinical audit data was collected and managed using Research Electronic Data Capture (REDCap) electronic data capture tools, a secure, web-based software platform hosted at Curtin University (Harris et al., 2019; Harris et al., 2009). A specific ‘Risk Factors’ instrument within the REDCap data collection form was designed by the authors (EH, AM, DG, LM) and created for collection of factors. The factors were sorted into different domains/categories in the REDCap instrument for ease of collection, and open text response questions were included in each domain/category allowing for collection of extra linked factors. Phase three was completed by one auditor (EH).

Data analysis

A convergent mixed methods approach was utilised (Pluye & Hong, 2014). The sample of 160 notes was used as this was the available sample provided by conducting companion research. The companion validity study included factor analysis and required twenty scores per item (eight items) for statistical analysis (Heron et al., 2021). The current sample of 160 allowed for adequate collection of linked factors for the development of this framework. For the qualitative analysis, sufficient information power was achieved by thematic saturation (Braun & Clarke, 2019; Hennink, Kaiser & Marconi, 2017; Tong, Sainsbury & Craig, 2007). This was done by determining specific study aims, sample specificity (mothers with ICLB), established theory (ICLB risk factors) and quality data collection, via use of a specific data collection form and the auditor holding expertise in women’s health physiotherapy and research (Malterud, Siersma & Guassora, 2016).

Descriptive statistics, counts and percentages, were calculated for the audit identified factors. This enhanced the development of the adapted model by highlighting factors considered important for inclusion. The qualitative process determined factor inclusion (see phase 5). The quantitative calculations were performed within the cleaned data file (Microsoft Excel) exported from REDCap, using the Microsoft Excel sum function (EH). With respect to socioeconomic status (SES) of the clinical audit population, four variables, postal area code, occupation, education, and private health insurance, were collected. Based on the amount of missing data, only postal area code was used to determine SES in this framework, using the Australian Bureau of Statistics Socio-Economic Indexes for Areas 2016, Index of Relative Socio-Economic Advantage and Disadvantage. The index ranks postal area codes within Australia from one (most disadvantaged) to 10 (most advantaged). A postal area index decile of six and above was considered high SES for this study, as determined by author consensus (EH, LM).

Qualitative analysis was required to synthesise the extra factors identified from the clinical audit, to identify the common theme and factor classification from the mother’s description of events linked with the onset of their ICLB episode. A combination of deductive and inductive coding was used to fit and assign the data to factors (EH) (Braun & Clarke, 2006). This qualitative/thematic process was independently cross-checked by another author (AM), to improve credibility of classification.

Confounding impacts, or linked factors potentially affecting the outcome of the proposed adapted model, were rationalised via the exclusion of these factors in phases 3 and 5, according to specified criteria (see Table S2). Qualitatively, rigor and strength of the research (trustworthiness) were established via the strategies outlined in Table 1, in accordance with the trustworthiness criteria for thematic analysis (Braun & Clarke, 2006; Nowell et al., 2017; Smith & McGannon, 2017).

Table 1 The thematic analysis process and means of establishing rigour and trustworthiness.

Thematic analysis phases	Strategies used to establish rigour and trustworthiness	
1. Data familiarisation	Data immersion
Triangulated data collected from different sources (the existing model, prior research, and clinical audit)
Documented initial thoughts and ideas for coding/themes
Raw data stored in well-organised Excel spreadsheets	
2. Initial code generation	Research team debriefing and triangulation (during research team meetings)
Utilised a coding framework
Established an audit trail (research team meeting minutes, code manual)
Documented all research team meetings (meeting minutes)	
3. Theme identification	Research team triangulation (during research team meetings)
Diagrammed to comprehend & visualise theme connections (using the prior model)
Documented development and organisation of codes and themes	
4. Theme refinement	Research team triangulation (during research team meetings)
Themes vetted by research team (during research team meetings)
Reviewed raw data to ensure themes reflective & grounded	
5. Theme finalisation (defining and renaming)	Research team debriefing and triangulation (during research team meetings)
Consensus on themes achieved amongst research team
Documented theme modification/organisation and naming (research team meeting minutes, diagrams)	
6. Report production	Research team debriefing (during research team meetings)
Sufficiently described process of coding and analysis
Rich context descriptions (including theme examples in tables)
Reported methodological, analytical, and theoretical rationale throughout paper	
Note:

Table adapted from Table 1, Nowell et al. (2017).

Phase 4: draft adapted model

A draft adapted model was created (EH), inclusive of all eligible factors that remained for consideration of inclusion into the adapted model after phase one to three. This model also comprised the quantitative and qualitative data from phase three.

Phase 5: proposed adapted model

The draft adapted model underwent factor fitting, integration, and modification to determine conceptual coherence for ICLB associated factors. An iterative process was employed, involving online collaborative research team meetings (EH, AM, MC, DG, LM). Changes were made to category, subcategory, and factor terminology to best represent ICLB. The research team drew upon theoretical constructs to help guide the categorisation. Clinical acumen was used to remove factors (n = 8) that had no strong physiological significance or rationale, and no underpinning logical or physiological mechanism (see Table S2). Some factors underwent minor amendment (grouping or renaming) to align with the presented structure of the proposed adapted model. Consensus on factor fitting, integration and preferred category, subcategory, and factor terminology was achieved amongst the multidisciplinary research team (EH, AM, MC, DG, LM) to produce the resultant proposed ‘ICLB reasoning model’. The research team members hold expertise in women’s health physiotherapy clinical practice and research, with a combined total of 93 and 46 years of expertise, respectively.

Results

Demographics

Mothers from the clinical audit had a median age of 35 years and were a median of 9.5 weeks postpartum at the time of their ICLB (see Table 2). Most mothers were primiparous, employed and of high socioeconomic status (see Table 2 and Fig. 2). Mothers had an average of 7.3 factors for their current ICLB episode. The more common factors included antibiotic use during breastfeeding, history of an ICLB, any breast pump use, birth interventions, decreased milk transfer, breastfeeding behaviour and practices, nipple pain and ‘fit and hold’ (Douglas & Keogh, 2017) (attachment and positioning) difficulty (see Fig. 2). These factors were all reported in the recent narrative synthesis on risk factors for lactational mastitis (Wilson, Woodd & Benova, 2020), indicating the clinical audit population are likely representative of mothers that present with mastitis.

Table 2 Clinical audit mothers’ demographics (n = 160).

Demographic variable	Median (Q1, Q3) or n (%)	
Maternal agea (years)	35 (31, 37)	
Range	25–42	
Maternal parityb		
Primiparity	85 (53.1%)	
Multiparityc	68 (42.5%)	
Two children	58 (36.3%)	
Three children	8 (5%)	
Singleton birth	160 (100%)	
Mode of deliveryd		
Vaginal	90 (56.3%)	
Caesarean	37 (23.1%)	
Socioeconomic status		
Postal area indexe		
1f	0 (0%)	
2	5 (3.1%)	
3	4 (2.5%)	
4	11 (6.9%)	
5	2 (1.3%)	
6	2 (1.3%)	
7	26 (16.3%)	
8	6 (3.8%)	
9	37 (23.1%)	
10g	59 (36.9%)	
Infant age (weeks)	9.5 (4.0, 21.7)	
Range	0.57–91.25	
Notes:

Q, Quartile.

a n = 2 maternal date of births not reported.

b n = 7 not reported.

c n = 2 not reported.

d n = 33 not reported.

e n = 8 not reported.

f Most disadvantaged.

g Most advantaged.

Data source: Heron et al. (2021).

Figure 2 ICLB reasoning model.

Adapted from: ‘Pain and Movement Reasoning Model’ (Jones & O’Shaughnessy, 2014) and ‘Breastfeeding Pain Reasoning Model’ (Amir, Jones & Buck, 2015). PHx = Past History. *Encompasses caesarean mode of delivery, induction, and preterm birth (defined as <37 weeks gestation). #’Fit and hold’ identified from (Douglas & Keogh, 2017). ^Encompasses nipple dressings/patches and nipple shields.

Development of the proposed clinical framework: ‘ICLB reasoning model’

The flow of factors through the study is shown in Fig. 1. Overall, 57 factors comprised the draft adapted model and were considered for inclusion into the adapted model. Of these factors, 13 remained from the existing model and 44 were identified collectively from the prior research and clinical audit data. The prior research factors not found in the clinical audit population (13) and thus not considered for inclusion were anaemia, not cleaning the nipple before breastfeeding, education level, water given to infant in first month, history of childhood sexual abuse, income, maternal throat infection, menstruation returned, multiples (twins/multiples), pacifier use, prelacteal feeding, Staphylococcus aureus isolated from nipple, infant and breast milk, and smoking.

Three research team meetings were required to achieve consensus on factor fitting, integration, and framework modification. The three categories on the apices of the existing model were maintained, with one minor terminology change to best reflect ICLB associated factors; ‘Local stimulation’ changed to ‘Local influences’. The resulting three categories in the proposed adapted ICLB model are: (1) CNS modulation, (2) external influences, (3) local influences. Consensus from the iterative process resulted in five terminology changes to the existing subcategories, and a total of 34 factors in the proposed adapted ICLB model after factor consolidation and terminology modification (Fig. 2 and Table S3). Only nine factors were added to the existing model (25 factors) (Fig. 1).

Lastly, the existing triangle model was converted into a tree diagram (EH) thus differentiating from the prior clinical reasoning models while symbolising the branching breast ductal system and the conceptual relationship between fertility and ‘tree of life’ (Becker, 2002). The proposed categories, subcategories and fitted factors of the adapted ‘ICLB reasoning model’ are outlined below.

CNS modulation

The first subcategory, ‘Afferent input’, was adapted from ‘Prolonged afferent input’ in the existing model, to incorporate acute as well as chronic afferent input. Nipple injury, a term to encompass all forms of nipple damage, which is frequented by pain, was moved to this subcategory from the ‘local stimulation’ category in the existing model. The ICLB factor, traumatic birth, was moved from the existing ‘pre-disposing factors’ category (previous trauma) and fitted to the second subcategory, ‘Cognitive – emotive – social state’, to join other factors that may impact a mother’s cognitive, emotive, or social state and have been linked with ICLB. The third subcategory, ‘Pre-existing’, was adapted from ‘Predisposing factors’ in the existing model, to represent pre-existing factors such as maternal demographics that are linked to ICLB. The overarching factor term ‘birth interventions’ was used in this subcategory to represent caesarean section, induction of labour and preterm birth; factors related to labour and delivery that may increase susceptibility to ICLB (Yin et al., 2020). Some factor examples from the clinical audit data for this category are provided in Table 3.

Table 3 CNS modulation.

Subcategory	Linked factors	Examples	
Afferent input	Nipple injury	Cracked or damaged nipple	
Cognitive – emotive – social state	Maternal health: Infection	A cold, gastroenteritis	
Pre-existing	Birth interventions	Cesarean section, induction, preterm birth	
	PHx ICLB	Blocked ducts, pathological engorgement, mastitis, breast abscess (current or prior lactation)	

External influences

The subcategory ‘Miscellaneous’, which comprises products that may directly or indirectly influence breast inflammation (Angelopoulou et al., 2018; Contreras & Rodríguez, 2011; Geddes et al., 2017), was removed from this category to stand alone in the trunk of the tree. Of the remaining three subcategories, ‘Physiological’ was added to the existing subcategory ‘attributes of mother’, to differentiate between the other maternal attributes in the adapted CNS modulation category. Within this adapted subcategory, the over-arching term ‘nipple anatomy’ was used to encompass all physiological variations in the maternal nipple that may contribute to ICLB (Yin et al., 2020). Similarly, in the ‘attributes of infant’ subcategory, oral anomaly was used to encompass tongue tie, small mouth and other anatomical anomalies reported in the clinical audit population that may influence ICLB. In the last subcategory, ‘interaction between mother and infant’, the factor term ‘fit and hold’ was chosen to represent suboptimal attachment and positioning of the baby at the breast, because the way a mother’s and infant’s anatomy fit together are important components in ensuring optimal milk transfer (Douglas & Keogh, 2017). Some factor examples from the clinical audit data for this category are provided in Table 4.

Table 4 External influences.

Subcategory	Linked factors	Examples	
Attributes of infant	Ill health	Blood in stools, a cold, ear infection, oral thrush, sickness, teething, upset tummy.	
	Oral anomaly	High palate, lip tie, retrognathic jaw, short tongue, small mouth, tongue tie.	
Interaction between mother and infant	Breastfeeding behaviour & practices	Consecutive same breast, duration >30 mins, feeding <6 times/day, feeding 6–9 times/day, feeding more frequently in 48 h before onset, position – lying down, sleeping with sucking, behaviour change – e.g., Christmas, introduced solids, trying to introduce a bottle.	
	Decreased milk transfer	Baby refusing breast, incomplete drainage, longer interval between feeds/expression, missed a feed, reduced length of feeds, weaning/tried to wean.	
Physiological attributes of mother	Nipple anatomy	Horizontal fold, inversion/retraction, short nipple.	
	PHx non-lactational breast complications	Breast cancer, breast surgery (benign mass removal), cysts, fibroadenoma.	
Miscellaneous	Antibiotic use	During this lactation period for; this ICLB, a prior ICLB, other infection.	
	External nipple contacts	Hydrogel dressings, nipple shield(s).	
	Nipple creams	Antifungal, lanolin, steroid cream.	

Local influences

The two existing subcategories of ‘chemical stimulation’ and ‘skin breakdown’ were changed to ‘stimulation’ and ‘breakdown’ respectively, in the adapted model, to better represent ICLB factors. ‘Stimulation’ signifies mechanical stimuli to the breast that may influence ICLB, such as external compression (e.g., from an ill-fitting bra) or firm self-massage. The subcategory ‘breakdown’ was retitled to capture abnormalities or local breast signs/symptoms/features, not normally present in the lactating breast (Geddes, 2007), that have been linked with ICLB. Some factor examples from the clinical audit data for this category are provided in Table 5.

Table 5 Local influences.

Subcategory	Linked factors	Examples	
Stimulation	External compression	Ill-fitting bra (tight, underwire), sleeping position (lateral, prone), baby kick/knock to breast, side of bassinet (while settling baby), baby sleeping on chest, electric pump flange.	

Discussion

The prior ‘Breastfeeding Pain Reasoning Model’ (Amir, Jones & Buck, 2015) for nipple pain was successfully adapted for the categorisation of ICLB linked factors, by the addition of nine factors across all nine subcategories. The resultant clinical framework, the ‘ICLB reasoning model’ (Fig. 2), is an encompassing tool that facilitates clinical assessment and treatment of ICLB. It is a valuable step in providing clarity around the many factors for ICLB, for both clinicians and researchers.

Clinicians treating mothers with ICLB require a framework that summarises the different sources of information they might use, to identify factors they need to consider when assessing and treating ICLB. Clinicians commonly refer to frameworks and models, search prior research, consult peers, and review past treatment notes, when determining factors that may influence a patient’s clinical presentation. This is time consuming and therefore, difficult as a working clinician, as evidenced by the recent COVID pandemic. Thus, this is an important study because it provides an encompassing adapted model to assist clinicians to consider all factors linked with ICLB, that are derived from many sources used to inform clinical practice. Notably for researchers, it emphasises the need for robust, prospective studies to investigate and clarify ICLB associated factors.

The reasoning models used for the basis of the ICLB model were designed to enable clinicians to consider the range of influencing factors, identify the predominant contributing factors for each patient and from this, select appropriate management techniques (Amir, Jones & Buck, 2015; Jones & O’Shaughnessy, 2014). The proposed ICLB model aims to do this for the linked factors of ICLB. This adaptation is supported by a recent qualitative study which found the ‘Pain and Movement Reasoning Model’ to be appropriate and beneficial across a range of physiotherapy clinical areas including women’s health and to have potential to be adapted for other multifactorial conditions (Jones et al., 2021). Thus, this adaptation requires further evaluation in wider settings, such as within the public health care system. The categories used in the existing pain models were transferable and applicable for categorisation of the broad range of ICLB factors. This suggests that factors shown to influence and modulate pain may similarly influence other conditions of tissue pathology, such as ICLB. More research on the underlying mechanism is required, given the exact aetiology of ICLB is unclear and lacks consensus.

The factors fitted to this model still require empirical examination. This model may therefore assist researchers by highlighting which prognostic or determining factors need to be considered for use in robust predictive modelling and casual research (Kent et al., 2020). Clinically, a broad approach or overall plan to examine ICLB associated factors is required. Thus, the broad categories and terminology used in this framework may be useful for clinicians and mothers pending high-quality research to confirm strength and direction of association. Furthermore, the impact of other maternal comorbid health conditions that may contribute to pain, on ICLB have not been considered in the prior research and thus were not included in the adapted model. Future larger studies would be needed to determine which health co-morbidities may predispose mothers to mastitis. Importantly, this proposed model is preliminary, and should be modified and updated based upon the findings of future research. It is specifically developed as a clinical management tool, designed to inform management and practice, by facilitating the clinician’s consideration of all potential factors that need to be addressed when treating mothers with ICLB. This tool has not currently been assessed as either a screening or predictive tool, and until this research has been undertaken the authors recommend that it should not be utilised for these purposes.

Limitations of this study include those common to research in this area, as outlined in previous studies (Kent et al., 2020; Wilson, Woodd & Benova, 2020). Specifically, the linked factors identified in the clinical audit, as well as those highlighted in the existing literature, are subject to recall bias and the direction of causality can be difficult to determine. Furthermore, due to the retrospective design of the clinical audit, if a factor was not documented in the clinical notes, it was assumed absent. Conversely, input from the clinical audit into the development of the framework brings strength to this study. It enabled the framework to be tailored for clinical utility and the target population of lactating mothers with ICLB thus ensuring clinical suitability. The clinical audit population was from only one private physiotherapy practice and of high socioeconomic status. Therefore, the linked factors may not be reflective of low socioeconomic populations. Another strength of this study was the research team. They brought ICLB clinical and research expertise to the development of the framework, which is based on an existing clinical reasoning model that had been adapted for breastfeeding by experts in the field and found to have clinical suitability and utility in women’s health physiotherapy. Of note, there are elements of the proposed adapted model that may need revision when future research on ICLB associated factors becomes available.

Translation to clinical practice

The proposed ‘ICLB reasoning model’ could be employed to consider the wide range of factors influencing ICLB and assist in targeted treatment selection. The clinician could complete the framework with the mother as part of the clinical interview, prompting discussion regarding management strategies and direction of education and advice. The tree diagram permits clinicians to visually determine the best representation of the contribution of each category, or major branch. Mothers can present with multiple associated factors, often across all three categories, with interdependence between categories and factors. Many of these factors are modifiable or remediable by conservative management strategies inclusive of the common support measures in the literature (Amir & The Academy of Breastfeeding Medicine Protocol Committee, 2014). There is a critical lack of high-level evidence regarding which ICLB linked factors must be considered and no high-level evidence for effective treatments for or prophylaxis of ICLB exist (Wilson, Woodd & Benova, 2020). Therefore, this model could prove useful to researchers designing and implementing future epidemiological and interventional studies in this area (Wilson, Woodd & Benova, 2020).

Conclusions

The proposed ‘ICLB reasoning model’ provides a thorough categorisation of potential ICLB related linked factors, to assist clinicians in identifying the contributors to a mother’s clinical presentation. This adapted model contains 34 factors, organised into three broad categories, (1) CNS modulation, (2) external influences, (3) local influences, and nine subcategories. By considering the range and interdependence of factors, clinicians can more effectively target management strategies, facilitating recovery from these often, debilitating conditions. There is a substantial need for high quality evidence for ICLB associated factors, which could enhance/strengthen this framework.

Supplemental Information

Supplemental Information 1 The ‘Breastfeeding Pain Reasoning Model’ factor categorisation (Amir, Jones & Buck, 2015).

Click here for additional data file.

Supplemental Information 2 Excluded variables across the phases of development of the ‘ICLB reasoning model’.

Click here for additional data file.

Supplemental Information 3 Factor fitting process.

Click here for additional data file.

Supplemental Information 4 Clinical audit data: Mothers’ demographics and ICLB linked factors.

The de-identified demographic data and ICLB linked factors collected in the retrospective clinical audit, which was used to calculate the counts and percentages in the adapted model.

Click here for additional data file.

We would like to acknowledge Lester Jones and Des O’Shaughnessy, co-creators of the Pain and Movement Reasoning Model, and Lisa Amir and Miranda Buck, who created the Breastfeeding Pain Reasoning Model which was used for adaptation of the ICLB framework. We would also like to thank Hillary Schwantzer, the director of the private physiotherapy practice where the clinical audit was performed.

Additional Information and Declarations

Competing Interests

Author Contributions

Human Ethics

Data Availability

Melinda Cooper is the owner and director of MMC Physiotherapy.

Emma Heron conceived and designed the experiments, performed the experiments, analyzed the data, prepared figures and/or tables, authored or reviewed drafts of the article, and approved the final draft.

Adelle McArdle conceived and designed the experiments, analyzed the data, authored or reviewed drafts of the article, and approved the final draft.

Melinda Cooper conceived and designed the experiments, authored or reviewed drafts of the article, and approved the final draft.

Donna Geddes conceived and designed the experiments, authored or reviewed drafts of the article, and approved the final draft.

Leanda McKenna conceived and designed the experiments, analyzed the data, prepared figures and/or tables, authored or reviewed drafts of the article, and approved the final draft.

The following information was supplied relating to ethical approvals (i.e., approving body and any reference numbers):

Ethics approval was obtained from Curtin University Human Research Ethics Committee (HRE2020-0544).

The following information was supplied regarding data availability:

The raw data is available in the Supplemental File.

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
