# Peer review of "Adaptation of a clinical reasoning model for use in inflammatory conditions of the lactating breast: a retrospective mixed-methods study"

_PeerJ, doi:10.7717/peerj.13627_

## Round 0.1 · original submission · Major Revisions

I have read the feedback from each of the reviewers and recommend that you revise your manuscript in line with their comments and the PeerJ editorial criteria: https://peerj.com/about/editorial-criteria/.

·

Basic reporting

No comments

Experimental design

1. In phase 5, the author demonstrated that an iterative process was employed to determine factor inclusion. However, there are no clear criteria (numbers or qualitative information) about how to determine whether a factor should be included or not. The author should provide statistics or qualitative statements about the qualification of individual factors in the adapted model.

2. Given that all participants are from a mono-center study (a private physiotherapy practice in Melbourne, Victoria), some factors cannot be simply determined as “linked factors” to ICLB. For example, the authors need to justify whether the high socioeconomic status in this study is because the postal code around Melbourne is more likely to be above 6. If so, the high socioeconomic status actually reflects the economic advantage of the Melbourne area but has nothing to do with ICLB.

Validity of the findings

I suggest the author compare their model with the existing Breastfeeding Pain Reasoning Model in an irrelevant dataset to see if the adapted model performed better than the BPRM on ICLB.

·

Basic reporting

This is a well-design study evaluating the contributing factors of ICLB. Authors have aggressively updated the existing model on nipple pain i.e. Breastfeeding Pain Reasoning Model (BPRM). However, the rationale of the study is quite weak, the authors have described the confounding impact in the existing model but did not describe the extent and nature of such confounding during the rational statement of the study. However, the authors have provided many details on the significance and importance of this study in the introduction section which I believe should be discussed in the discussion section.

Experimental design

This study is well-designed and controlled in all aspects. However, there are a few things that should be considered in the method section. Authors should put a heading of operational definition and define the terms used throughout the manuscript.
The authors have discussed sample size estimation but did not describe the formula or equation used to estimate the sample. It's not clear what prevalence of ICLB was used to estimate the sample size. I am more concerned about the size as it is quite low. However, I agree that being a mono-centric study, a low sample size remains a barrier.
Consultation from physiotherapists is not common in various developing countries, it is needed to provide information on whether all patients with nipple pain are attended by physiotherapists at the study location. Moreover, it is suggested to please add more information about the study location in the method section.
I suggest describing the search strategies adopted to find the existing data on this topic as this data is used in the analysis and modeling.

Validity of the findings

The findings and results of this study are well presented in an adequate number of figures and tables. I have a query here; most of the demographic data are related to pregnancy outcomes. Can the authors explain why other social and clinical variables were not collected during the data collection process? For example, certain comorbid conditions may also affect the ICLB and their association with it is of utmost importance, particularly in endocrine disorders. Some other factors i.e. abnormal menstrual cycle, hormone replacement therapy, etc. may also contribute to pain. I am just suggesting considering the impact of these conditions on the model, as they may also confound the findings. I think the authors can discuss the same in the discussion section.
The authors have not discussed the low sample size as a limitation of this study.
The discussion section is very brief (hardly a few sentences). The authors did not discuss the relationship of the factors with ICLB. I can understand, the author cannot discuss each factor here but at least there should be information on how identified factors related to pain i.e. a brief plausible mechanism.

Additional comments

The conclusion is very general. Authors should state the number of factors included in the model along with the broad categorization of these factors.

---

## Round 0.2 · accepted · Accept

The authors have addressed all comments from the reviewers in an appropriate way.

·

Basic reporting

The manuscript describes a management tool of ICLB. This will largely benefit both clinicians and patients. The manuscript is well written in professional English. The data provided in the manuscript explain the hypotheses clearly.

The author addressed all questions from the reviewer in an appropriate way.

Experimental design

The author provided modifications addressing the question from a reviewer.

Validity of the findings

The author provided additional explanations for the potential application of the ICLB tool and clearly stated the aspects that can be further improved in this ICLB tool.

·

Basic reporting

I have no more comments on the manuscript.

Experimental design

I have no more comments on the manuscript.

Validity of the findings

I have no more comments on the manuscript.

Additional comments

I have no more comments on the manuscript.